# *Babesia bovis* Enolase Is Expressed in Intracellular Merozoites and Contains B-Cell Epitopes That Induce Neutralizing Antibodies In Vitro

**DOI:** 10.3390/vaccines13080818

**Published:** 2025-07-31

**Authors:** Alma Cárdenas-Flores, Minerva Camacho-Nuez, Massaro W. Ueti, Mario Hidalgo-Ruiz, Angelina Rodríguez-Torres, Diego Josimar Hernández-Silva, José Guadalupe Gómez-Soto, Masahito Asada, Shin-ichiro Kawazu, Alma R. Tamayo-Sosa, Rocío Alejandra Ruiz-Manzano, Juan Mosqueda

**Affiliations:** 1Immunology and Vaccines Laboratory, Faculty of Natural Sciences, Autonomous University of Queretaro, Santiago de Queretaro 76230, Queretaro, Mexico; alma.cardenas@uaq.mx (A.C.-F.); diego.hernandez@uaq.mx (D.J.H.-S.); 2Ph.D. Program in Biological Sciences, College of Natural Sciences, Autonomous University of Queretaro, Santiago de Queretaro 76230, Queretaro, Mexico; 3Posgrado en Ciencias Genomicas, Universidad Autonoma de la Ciudad de Mexico, Mexico City 03100, Mexico; minerva.camacho.nuez@uacm.edu.mx; 4Animal Diseases Research Unit, USDA-ARS, Pullman, WA 99164, USA; massaro.ueti@usda.gov; 5College of Veterinary Medicine, Autonomous University of Chiapas, Tuxtla Gutierrez 29000, Chiapas, Mexico; mario.hidalgo@unach.mx; 6Cuerpo Academico de Salud Animal y Microbiología Ambiental, Natural Sciences College, Autonomous University of Queretaro, Santiago de Queretaro 76230, Queretaro, Mexico; angelina@uaq.mx; 7Cuerpo Académico de Nutrición y Reproducción Animal, Facultad de Ciencias Naturales, Autonomous University of Queretaro, Santiago de Queretaro 76230, Queretaro, Mexico; jose.gomez@uaq.mx; 8National Research Center for Protozoan Diseases, Obihiro University of Agriculture and Veterinary Medicine, Obihiro 080-8555, Hokkaido, Japan; masada@obihiro.ac.jp (M.A.); skawazu@obihiro.ac.jp (S.-i.K.); 9Instituto de Investigaciones en Ciencias Veterinarias, Universidad Autónoma de Baja California, Mexicali 21386, Baja California, Mexico; almatamayo@uabc.edu.mx; 10Departamento de Biología Celular y Fisiología, Instituto de Investigaciones Biomédicas, Universidad Nacional Autónoma de Mexico, Mexico City 04510, Mexico; rocio_ruiz@iibiomedicas.unam.mx

**Keywords:** *Babesia bovis*, enolase, synthetic peptide, B-cell epitope, bioinformatics, in vitro culture

## Abstract

**Background**: Bovine babesiosis, caused by the tick-borne apicomplexan parasite *Babesia* spp., is an economically significant disease that threatens the cattle industry worldwide. *Babesia bovis* is the most pathogenic species, leading to high morbidity and mortality in infected animals. One promising approach to vaccination against bovine babesiosis involves the use of multiple protective antigens, offering advantages over traditional live-attenuated vaccines. Tools such as immunobioinformatics and reverse vaccinology have facilitated the identification of novel antigens. Enolase, a “moonlighting” enzyme of the glycolytic pathway with demonstrated vaccine potential in other pathogens, has not yet been studied in *B. bovis*. **Methods**: In this study, the enolase gene from two *B. bovis* isolates was successfully identified and sequenced. The gene, consisting of 1366 base pairs, encodes a predicted protein of 438 amino acids. Its expression in intraerythrocytic parasites was confirmed by RT-PCR. Two peptides containing predicted B-cell epitopes were synthesized and used to immunize rabbits. Hyperimmune sera were then analyzed by ELISA, confocal microscopy, Western blot, and an in vitro neutralization assay. **Results**: The hyperimmune sera showed high antibody titers, reaching up to 1:256,000. Specific antibodies recognized intraerythrocytic merozoites by confocal microscopy and bound to a ~47 kDa protein in erythrocytic cultures of *B. bovis* as detected by Western blot. In the neutralization assay, antibodies raised against peptide 1 had no observable effect, whereas those targeting peptide 2 significantly reduced parasitemia by 71.99%. **Conclusions**: These results suggest that *B. bovis* enolase contains B-cell epitopes capable of inducing neutralizing antibodies and may play a role in parasite–host interactions. Enolase is therefore a promising candidate for further exploration as a vaccine antigen. Nonetheless, additional experimental studies are needed to fully elucidate its biological function and validate its vaccine potential.

## 1. Introduction

Bovine babesiosis is one of the most important tick-borne diseases of cattle worldwide. In Mexico, the disease has been reported since 1905, as the country’s environment provides favorable conditions for the survival of the tick that transmits it. In Latin America, this disease is transmitted by the cattle fever ticks *Rhipicephalus microplus* and *R. annulatus* [1,2]. The estimated annual economic losses caused by the cattle fever tick and the diseases they transmit, such as babesiosis and anaplasmosis, are between USD 13.9 billion and USD 18.7 billion [3]. Developing effective and sustainable methods to control this disease should be an important goal for research in countries where bovine babesiosis is endemic. Since the late 20th century, efforts have been made to develop an effective vaccine against bovine babesiosis. This has led to the creation of vaccines using attenuated live parasites, which have proven very effective. However, their use comes with several drawbacks, including the risk of reversion to virulence, high production costs, and potential contamination with other pathogens, among others [2,4].

Although there are no commercial vaccines without these disadvantages, to develop them it is necessary to deepen the understanding of the antigens and the mechanisms involved in the life cycle of pathogens that cause bovine babesiosis, both in the vector that transmits them, and in the mammalian host. Thus, the use of several tools is required, such as immunobioinformatics and reverse vaccinomics, among others, which have advanced significantly in recent years and make it possible to in silico identify genes that encode proteins capable of inducing antibody production to neutralize *B. bovis* invasion of its target cells [5,6].

Enolase is a glycolytic enzyme classified as a “moonlight” protein because it has been found to perform functions beyond its role in glycolysis. On the surface of certain pathogens, including *Plasmodium*, enolase facilitates host-cell invasion by binding to plasminogen; this interaction between enolase and plasminogen leads the formation of active plasmin which degrades extracellular matrix surrounding the target cell, facilitating pathogen entry [7,8]. It is considered a vaccine candidate for different diseases, including those caused by Apicomplexan organisms such as *Babesia microti* and *Plasmodium falciparum*, as it induces a protective immune response in mice immunized with recombinant enolase [7,8,9]. To date, enolase has not been characterized in *B. bovis*. This study aimed to identify the enolase gene in the *B. bovis* genome and to determine its expression in parasite erythrocytic stages, as well as to determine if the enolase protein contained conserved B-cell epitopes that neutralize parasite invasion in the bovine erythrocytes.

## 2. Materials and Methods

This project was evaluated and approved by the Natural Science Faculty Bioethical Committee from the Autonomous University of Queretaro (approval 14FCN2019).

### 2.1. Identification and In Silico Analysis of the B. bovis Enolase Gene

Using the *enolase* gene sequences of *Babesia bigemina* (NCBI accession code: MK490919.1) and *P. falciparum* (NCBI accession code: XM_004221995.1) as reference, we searched for similar sequences in the *B. bovis* strain T2Bo genomic sequence (NCBI accession code: AAXT00000000.2) within the GenBank database (https://www.ncbi.nlm.nih.gov/genbank, (accessed on July 2019) through a BLAST (version 2.10.0) analysis. For the matched sequences, we performed the following analyses: (1) prediction of open reading frames, which was carried out with the ORFfinder tool (https://www.ncbi.nlm.nih.gov/orffinder, accessed on July 2019). (2) Prediction of introns and exons, which was performed using the GENSCAN tool (http://hollywood.mit.edu/GENSCAN.html, accessed on July 2019).

### 2.2. B. bovis DNA Extraction from Field Isolates

The *B. bovis* isolates used in this project come from *R. microplus* ticks collected from cattle in three different and geographically distant states in Mexico: Nayarit, Michoacan, and Puebla. The ticks were processed at the Immunology and Vaccines Laboratory of the Autonomous University of Queretaro. Each tick was incubated individually for oviposition and then examined for *Babesia* spp. infection. The infection was detected by microscopically analyzing the hemolymph for the presence of kinetes [10]. DNA was extracted from kinete-positive ticks with the DNeasy Blood & Tissue kit (QIAGEN, Hilden, Germany), following the protocol specified by the manufacturer. A nested PCR analysis was performed to identify those ticks infected with *B. bovis* using the oligonucleotides described previously [11] and MyTaq^TM^ Mix (Bioline, Luckenwalde, Germany). The PCR thermal conditions for the first reaction were 95 °C for 3 min as initial denaturation, followed by 30 cycles at 95 °C for 30 s, 52 °C for 30 s and 72 °C for 30 s, and a final extension step of 72 °C for 7 min. A measurement of 1 µL of this reaction were used as a DNA template for the nested reaction. The thermal conditions for this second reaction were 95 °C for 3 min as initial denaturation, followed by 30 cycles at 95 °C for 30 s, 54 °C for 30 s, and 72 °C for 30 s, and a final extension step of 72 °C for 7 min. Positive DNA was preserved at −20 °C until use.

### 2.3. B.bovis RNA and Antigen Extraction

RNA and protein from *B. bovis* were obtained from the blood of an experimentally infected Holstein steer. An 8-month-old steer, sourced from a tick-free zone, was infested with approximately 10,000 larvae derived from *B. bovis*-infected female ticks of the Puebla state isolate. The blood of the infected steer was processed following the methodology described by Vega and others [12] to establish an in vitro culture to obtain only infected erythrocytes for slides and protein extraction. These erythrocytes were used for RNA extraction with TRIzol Reagent (Invitrogen, Carlsbad, CA, USA) using the protocol specified by the manufacturer. Additionally, infected erythrocytes were separated and used for protein extraction following the methodology described previously [13]. Blood smears were prepared as antigen for the indirect immunofluorescence test, following the indications of the WOAH manual of diagnostic tests and vaccines for terrestrial animals [14]. This material was preserved at −20 °C until use.

### 2.4. Amplification and Sequencing of Enolase Gene

Two oligonucleotides were designed to amplify the full length of the enolase gene (Fw: 5′-CACTTGTCCGCTCAGTCACTC-3′, Rv: 5′-CCCTGCGATACTGGAGTTTTG-3′) and were used in PCR assays with Nayarit and Michoacan DNA strains, using MyTaq^TM^ Mix (Bioline, Luckenwalde, Germany). The thermocycling conditions were as follows: an initial denaturation step at 95 °C for 3 min, followed by 35 cycles at 95 °C for 15 s, 53 °C for 30 s, and 72 °C for 1 min, with a final extension step at 72 °C for 7 min. The PCR product was visualized in 1% agarose gel with fluorescent GelRed^TM^ Nucleic Acid Gel Stain (GOLDBIO, St. Louis, MO, USA) and identified with the molecular size marker HyperLadder^TM^ 1 Kb (Bioline, Luckenwalde, Germany).

The amplicon obtained was purified using the EZ-10 Spin Column DNA gel Extraction MiniPrep kit (Bio Basic Inc., Markham, ON, Canada). The purified amplicons were cloned into the TOPO^TM^ TA Cloning^TM^ with pCR^TM^2.1-TOPO^TM^ One Shot TOP10 Chemically Competent *E. coli*^TM^ (Invitrogen, Carlsbad, CA, USA) according to the manufacturer instructions. The plasmidic DNA from selected clones (one from Nayarit strain and two from Michoacan strain) were purified using Illustra^TM^ plasmidPrep Mini Spin Kit (GE Healthcare, Buckinghamshire, UK) and sent for sequencing to Instituto de Biotecnologia at the Universidad Nacional Autonoma de Mexico (Cuernavaca, Morelos, Mexico) using the oligonucleotides designed to this purpose: 5′-CATGCTCCCAGTTCCATGC-3′ and 5′-GGTGTATGCTTCCCAGTCATCC-3′, and the general oligonucleotides “M13/pUC Reverse” and “T7 primer”. The obtained electropherograms were analyzed with the help of the NUCLEICS BioEdit program (https://www.nucleics.com) and were assembled to obtain the sequences corresponding to the *B. bovis* enolase gene from Nayarit and Michoacan isolates.

### 2.5. Transcription Analysis by RT-PCR

To evaluate the enolase gene transcription in erythrocytic phases, an RT-PCR assay was performed using the Super Script^®^ III Firts-Strand kit (Invitrogen, Carlsbad, CA, USA) following the manufacturer’s specifications and RNA from *B. bovis* Puebla state isolate. The cDNA obtained from the reverse transcription was used in a 12.5 µL PCR reaction composed as follows: 6.25 µL of MyTaq^TM^ Mix (Bioline, Luckenwalde, Germany), 5 µL of each of the oligonucleotides that amplify the internal part of the gene (Fw: 5′-CATGCTCCCAGTTCCATGC-3′ and Rv:5′-GGTGTATGCTTCCCAGTCATCC-3′) at 10 pmol/µL, 3.25 µL of molecular biology grade water, and 2 µL of cDNA. Molecular biology grade water instead of DNA or cDNA were used as a reaction negative control. *B. bovis* DNA was used as positive control and Dnase-traeated RNA from RT-PCR reaction in absence of Retro transcriptase was used as control for DNA contamination. The thermocycling program corresponds to the characteristics of the oligos used. The reaction product was visualized on a 1% agarose gel with GelRed^TM^ (GOLDBIO, St. Louis, MO, USA) and the molecular size marker Hyperladder^TM^ 100 BP (Bioline, Luckenwalde, Germany) was used as a reference.

### 2.6. Enolase Protein In Silico Analysis and B-Cell Epitopes Prediction

The *enolase* nucleotide sequences were obtained from isolates in the Mexican states of Nayarit and Michoacan. The predicted sequence from T2Bo strain were analyzed with the following bioinformatics tools: (1) GENESCAN (http://hollywood.mit.edu/GENSCAN.html, accessed on July 2019) was used for identification of introns and exons; (2) ORF Finder (https://www.ncbi.nlm.nih.gov/orffinder, accessed on July 2019) was used for open reading frame identification and to obtain predicted amino acid sequences; (3) SignalP (http://www.cbs.dtu.dk/services/SignalP, accessed on July 2019) was used for signal peptide identification in the amino acid sequence; (4) TMHMM (http://www.cbs.dtu.dk/services/TMHMM/, accessed on July 2019) was used for the identification of transmembrane regions; (5) SMART (http://smart.embl-heidelberg.de, accessed on July 2019) was used for protein domain identification; and (6) PROSITE (https://prosite.expasy.org) allowed us to identify functional sites. The isoelectric point and molecular weight were predicted with the help of the ComputepI/Mw tool of ExPASy (https://web.expasy.org/compute_pi/, accessed on July 2019). Additionally, the amino acid sequences obtained were compared with the *B. bigemina* enolase (QEG79394.1) and the sequence obtained from the genomic T2Bo sequence by making alignments with Clustal W (https://www.genome.jp/tools-bin/clustalw, accessed on July 2019). The percentage of identity between them was determined with the Clustal omega tool (https://www.ebi.ac.uk/jdispatcher/msa/clustalo, accessed on July 2019). To search for peptides with B-cell epitopes, the following bioinformatics tools were used: ABCpred (http://crdd.osdd.net/raghava/abcpred, accessed on July 2019), BCEpred (http://crdd.osdd.net/raghava/bcepred, accessed on July 2019), BepiPred (http://www.cbs.dtu.dk/services/BepiPred, accessed on July 2019) and IEDB (https://www.iedb.org). Two selected peptides were commercially synthesized in an 8-branches peptide system (MAPS-8) by Peptide 2.0 Inc. (Chantilly, VA, USA).

### 2.7. Generation and Titration of Anti-Enolase Antibodies

To determine the immunogenicity of the selected peptides, 8-week-old rabbits were used, which were maintained and handled in the facilities of the Autonomous University of Querétaro. Each synthetic peptide was solubilized with PBS pH 7.4 and emulsified in Montanide ISA 71 VG adjuvant (Seppic, Paris, France) at a concentration of 100 μg/mL. Two rabbits were immunized with each peptide, rabbits “5” and “16” were immunized with the peptide ENOL-1 and rabbits “9” and “15” were immunized with the peptide ENOL-2. Rabbit “2” was immunized with adjuvant as a control. Five immunizations were performed subcutaneously in the scapular region. Immunizations were repeated 5 times, with intervals of 21 days between each immunization. Blood samples in anticoagulant-free tubes were obtained from the auricular vein before each immunization. Fifteen days after the last immunization the final bleeding was obtained. The serum was separated by centrifugation at 3500 rpm for 5 min on a Science Med clinic centrifuge (Helsinki, Finland), and aliquoted into 1 mL volumes and stored at −20 °C until use. To evaluate the antibody titters on sera, an indirect-ELISA was performed using 96-well Polystyrene High Bind Microplates (Corning Inc., Corning, NY, USA) coated with 2 µg/mL of each peptide (100 µL each well) as described previously [13]. After overnight incubation at 4 °C the plates were blocked using 200 µL PBS with 5% skim milk (*w*/*v*). Pre-immunization sera and the final sera were tested in triplicates of serial double dilutions starting at 1:500 and ending at 1:1,024,000. Sera were incubated for 1 h, at 37 °C. Then 100 µL peroxidase-bound, goat anti-rabbit Ig (Jackson ImmunoResearch Laboratories, West Grove, PA, USA) was added and incubated for 1 h, at 37 °C, 200 rpm. The reaction was detected using 100 µL of substrate solution containing O-phenylenediamine dihydrochloride and hydrogen peroxide suspended in sodium citrate citric acid 0.1 M. The plates were read in an ELISA reader with an O.D at 450 nm. The average O.D. values of the pre- and post-immunization triplicates were analyzed with a *t*-test (*p* < 0.05).

### 2.8. Evaluation of B. bovis Enolase Expression

To evaluate the expression of *B. bovis* enolase, confocal microscopy and Western blot assays were performed following the methodology described previously [15]. For immunofluorescence, fresh antigen obtained from an established culture of *B. bovis* fixed on slides with acetone–methanol (90–10%) was used. The slides were blocked with 5% goat serum in PBS-Tween 0.2% and then were incubated with the rabbit anti-enolase sera at a dilution of 1:150 in PBS and incubated for 1 h at 37 °C on a humidity chamber, then were washed 3 times with 0.1% PBS-T for 5 min. After washing, the slides were allowed to dry at room temperature. A second incubation was performed with a goat anti-rabbit IgG antibody coupled with Alexa-488 (Thermo Scientific, Waltham, MA, USA) diluted 1:200 in PBS-T containing Hoechst 33,342 (Thermo Scientific, Waltham, MA, USA) for 1 h at 37 °C, followed by ten washes with PBS-T. Slides were analyzed in a confocal microscope (Leica TCS SP5 Confocal Laser Scanning Microscope) using specific lasers for Alexa-488, Hoechst 33,342 and brightfield. Images were processed and merged with the LAS Advanced Fluorescence software (version 2.7.3.9723) (Leica, Wetzlar, Germany).

For Western blot, 133 µg of *B. bovis* (Puebla state isolate) infected erythrocytes protein extracts and 5 µL molecular weight marker Opti-Protein Ultra Marker-G623 (Applied Biological Materials Inc., Richmond, BC, Canada) were separated on a 12% SDS-Poliacrylamide gel and transferred to a nitrocellulose membrane (Bio-Rad, Hercules, CA, USA). The membrane was blocked overnight at 4 °C and 200 rpm with 5% skim milk on TBS and cut into 0.5 cm strips. The strips were incubated individually for 1 h at 4 °C and 200 rpm with the specific antibodies at a concentration of 1:10,000, then the membrane trips were washed 3 times with TBS-T 0.05% for 5 min at 200 rpm and blocked overnight. After the second blocking step, the strips were incubated for 1 h in the same conditions with 1:15,000 anti-rabbit IgG (H + L) HRP conjugated antibody (Jackson ImmunoResearch Laboratories, West Grove, PA, USA) diluted in 0.05% TBS-T with 2% skim milk and washed 3 times. The strips were developed with the ECL^TM^ Western blotting Detection Reagents kit (Cytiva^®^, Marlborough, MA, USA) on ChemiDoc Imaging System (Biorad, Hercules, CA, USA).

### 2.9. Neutralization Assay

To determine if the anti-Enolase antibodies block the invasion of merozoites into erythrocytes, a neutralization assay was carried out using an in vitro culture of *B. bovis* as described previously [15]. In vitro culture of *B. bovis* T3Bo strain (provided by the ADRU-USDA laboratory at Washington State University) was maintained in 48-well plates, at 5% hematocrit in PL media (100 mL = pH 7.2; 29 mL F-12K Nutrient Mixture + L-glutamine Life Technologies, 29 mL Stable Cell IMDM Sigma Aldrich (St. Louis, MO, USA), 2 mL 0.5 M TAPSO Sigma Aldrich, 0.5 mL Antibiotic Antimycotic solution 100× Sigma Aldrich, 1 mM calcium chloride Sigma Aldrich, 100 μL Antioxidant Supplement 1000× Sigma Aldrich, 1 mL Insulin-Transferrin-Selenium-Ethanolamine 100× Sigma Aldrich, 1 mL 50% Glucose Teknova, 500 μL L-glutamine 200 mM GIBCO) supplemented with 40% normal bovine serum [16], initially inoculated with 1% parasitemia, keeping it in an incubator at 37 °C at 5% CO_2_. Daily, 75% of the medium was replaced with fresh medium and parasitemia was counted. Every 2 days a subculture was performed to adjust the culture again to 1% parasitemia. The assay was performed in triplicate in a 96-well plate with a final volume of 200 µL per well under the abovementioned conditions, adding 1:10 of the serum from each rabbit. All antisera used in this experiment were decomplemented before use by heating at 56 °C for 30 min. After 72 h of incubation, each well was re-homogenized by pipetting and individual samples were taken for smears, stained with Giemsa, and ~2000 cells from each well were counted to obtain the percentage of parasitized erythrocytes (PEP). As a statistical analysis, T tests for independent samples were performed, where *p* values < 0.05 were considered significant.

## 3. Results

### 3.1. B. bovis Contains an Enolase Gene That Is Transcribed in Infected Erythrocytes

Bioinformatic analysis allowed the identification of the *enolase* gene on chromosome 3 of the *B. bovis* T2Bo strain genome, which is located at the locus BBOV_III007950 and is flanked by two genes encoding hypothetical proteins (loci BBOV_III007940 and BBOV_III007960). The gene has a size of 1366 bp, has a single intron that starts at position 40, and ends at position 94 (Figure 1A). The *B. bovis* enolase gene has an identity with its homolog in *B. bigemina* of 76.28% and 62.76% with *P. falciparum* enolase (Appendix A).

The reverse transcription-PCR analysis of cDNA obtained from IEP generated an amplicon of the expected size (Figure 1C).

The complete nucleotide sequences of the enolase gene from the Nayarit and Michoacan isolates were obtained, and the predicted amino acid sequences were each 442 residues in length. The predicted molecular weight of the enolase protein is 47.73 kDa and its isoelectric point is 5.98. Enolase was predicted to have no signal peptide or transmembrane regions. Protein domain analysis indicated the presence of previously reported enolase domains as “Enolase_N”, starting at position 4 and ending at position 139, “Enolase_C” starting at position 149 and ending at position 444. Two peptides with predicted B-cell epitopes were identified and named ENOL-1 and ENOL-2 (Figure 1B). Also, the signature domain of enolases was found in the sequence from 349 to 362. Enolase of the Nayarit isolate has an identity of 98.64% with the same T2Bo strain protein, while the identity of the isolate of Michoacan with T2Bo strain protein is 99.77% and with Nayarit isolate is 98.87% (Figure 2).

### 3.2. Enolase Peptides Contain Conserved, Immunogenic, B-Cell Epitopes

The ELISA test results indicated that antibodies were generated against each of the peptides from the first immunization at 1:2000 dilution (Appendix A). Fifth and last immunization to rabbits “5” and “16” with ENOL-1 yielded antibody titers of 1:128,000 and 1:256,000, respectively, and rabbits “9” and “15” immunized with ENOL-2 yielded antibody titers of 1:512,000. Rabbit “2” (control) showed no antibody titers against any of the peptides (Appendix A).

### 3.3. Enolase Gene Is Expressed in Erythrocyte Phases of B. bovis

Indirect immunofluorescence assay was performed, and the results are shown in Figure 3. The pre-immunization serum of rabbit “5” immunized with ENOL-1 did not recognize the parasites as can be seen in panels A–D. Panels E-H shows the results obtained with the post-immunization serum of rabbit “5” immunized with ENOL-1: antibodies detected the protein enolase, which is observed in green in the cytoplasm of the parasite and in sections of the membrane a slightly more intense signal. Panels I-L correspond to the pre-immunization serum of rabbit “15” immunized with ENOL-2, while panels M-P shows the results obtained with the post-immunization serum of rabbit “15” immunized with ENOL-2, the protein enolase was detected in the cytoplasm. Panels Q-T correspond to IEP incubated with the control serum of a rabbit immunized with adjuvant, where no signal was detected.

A Western blot was performed with lysates from a *B. bovis*-infected erythrocytes. The antibodies from rabbits immunized with peptides ENOL-1 and ENOL-2 recognized a protein of approximately 47 Kda which corresponds to the predicted weight of the enolase protein (Figure 4A). No band was recognized with pre-immunization sera not serum of a rabbit immunized with adjuvant (Figure 4A).

### 3.4. Antibodies Against B. bovis Enolase Block Parasite Invasion In Vitro

To demonstrate that anti-enolase antibodies have biological activity in a *B. bovis* culture, a neutralization assay was performed using the serums of rabbits immunized with ENOL-1 and ENOL-2 peptides and control. The results showed a statistically significant difference (*p* < 0.05) of percentage of parasitized erythrocytes (PPE%) only between the culture containing post-immunization serum of rabbits immunized with the peptide ENOL-2. There is no statistically significant difference between all pre-immunization serums and post-immunization serums of rabbits immunized with the peptide ENOL-1 and Control (Figure 4B).

## 4. Discussion

Bovine babesiosis stands out as one of the economically impactful tick-transmitted diseases in the livestock industry. The disease caused by *B. bovis* is of particular concern due to its high mortality rate. Currently, there are no recombinant vaccines available for this disease. Enolase, a moonlight protein present in various pathogens, has been considered a potential vaccine candidate. Although it is known to be present in other *Babesia* species, its experimental confirmation in *B. bovis* was lacking. Thus, the initial objectives of this project were to demonstrate the gene’s presence, assess its transcription and to determine if this antigen contained conserved B-cell epitopes that were immunogenic and finally, to test the ability of specific antibodies to block merozoite invasion in vitro.

Gene sequencing in two distinct *B. bovis* isolates, followed by bioinformatic analysis, revealed a 1366 bp enolase gene located on chromosome 3. The gene contains an intron, starting at position 40 and ending at position 94. This gene encodes a 422 amino acid protein with an approximate weight of 47.73 kDa. Unlike the findings by Liu and others [9] *B. microti* enolase is reported as a 1317 bp gene, coding for a 438 amino acid, 47.5 kDa soluble protein with no signal peptide or transmembrane regions, our results align with their general observations. Enolase, vital for glucose-dependent energy metabolism in various organisms, has been observed not only in the cytoplasm but also on the surface of eukaryotic cells, acting as a strong plasminogen-binding receptor [17,18]. The mechanism of enolase translocation to the cell surface remains unclear, lacking a signal peptide for membrane targeting, but its surface expression is crucial for fibrinolysis and intracellular invasion by various pathogens [17,19].

Comparative analyses indicated significant similarities between *B. bovis* enolase and that of another apicomplexan organism, *P. falciparum*, with a 1342 bp gene coding for a 446 amino acid and 48.7 kDa protein [9,20]. Despite *B. bovis* enolase having a slightly larger gene than its homologs in *P. falciparum* and *B. microti*, the encoded protein has fewer amino acids, possibly due to the differences in an intron in their *enolase* genes, as predicted by previous studies. Sequence alignments revealed high identity (98.64% and 99.77%) between the predicted enolase protein of the T2Bo strain and isolates from Nayarit and Michoacán, respectively. Additionally, the predicted enolase protein of *B. bigemina* showed identity percentages of 87.78% and 88.24% with those of *B. bovis* isolates from Nayarit and Michoacán, respectively.

The expression of enolase protein in *B. bovis* merozoites was assessed through confocal microscopy and WB. Specific antibodies against peptides ENOL-1 and ENOL-2, designed based on predicted antigenic regions containing predicted B-cell epitopes, successfully identified *B. bovis* in its intraerythrocytic stages. These results are comparable with those obtained by immunizing individuals with peptides from different proteins such as TRAP, RON-2 and AMA-1 of *B. bovis* and MIC-1 of *B. bigemina*, where similar labeling patterns were observed [13,21,22]. However, more detailed experiments are considered necessary to better determine the subcellular localization of enolase and whether it can be found on the surface of infected erythrocytes. However, verifying whether this occurs in natural infections would be difficult because erythrocytes infected with *B. bovis* tend to be sequestered in capillaries of important organs. Previous investigations also assessed synthetic peptide capacity to generate antibodies that neutralize *Babesia* invasion of erythrocytes, suggesting that peptides identified in this study may have the potential to inhibit *B. bovis* invasion, given similar strategies in their selection [13,15,22]. However, of the two peptides selected with the same strategy, and despite both being immunogenic, only one generated neutralizing antibodies. This may be due to different causes, among them we can hypothesize that the region in which the peptide is located is hidden in the native protein, which makes access to specific antibodies difficult, and another theory may be that the region from which the peptide was selected, although immunogenic, is not crucial for the function of the protein, so the antibodies generated would not represent an alteration in its function [23]. The fact that anti-enolase antibodies are not able to completely block the parasite’s ability to invade erythrocytes was expected, since other studies have observed that, despite completely blocking the activity of an antigen, by knocking out the gene that encodes it, the parasite manages to invade its target cell, suggesting that pathogens have more than one way to achieve invasion [24]. Additionally, we highlight the need to verify, in future experiments, the role that anti-enolase antibodies play in affecting infection in ticks fed on cattle immunized with this antigen.

## 5. Conclusions

*B. bovis* enolase is an active gene expressed in erythrocytic stages. This gene and the protein it encodes are highly conserved between different apicomplexan parasites and *Babesia* species. The *B. bovis* enolase protein contains two peptides with B-cell epitopes that are immunogenic, but only one ENOL-2 generates antibodies capable of neutralizing parasite invasion of erythrocytes in vitro culture. Therefore, the ENOL-2 peptide represents a promising vaccine candidate against *B. bovis* and should be further evaluated in future experiments, either alone or in combination with other epitope sequences. These studies should assess its ability to control infection by a virulent parasite isolate in vivo and, additionally, its impact on the arthropod vector.

## Figures and Tables

**Figure 1 vaccines-13-00818-f001:**
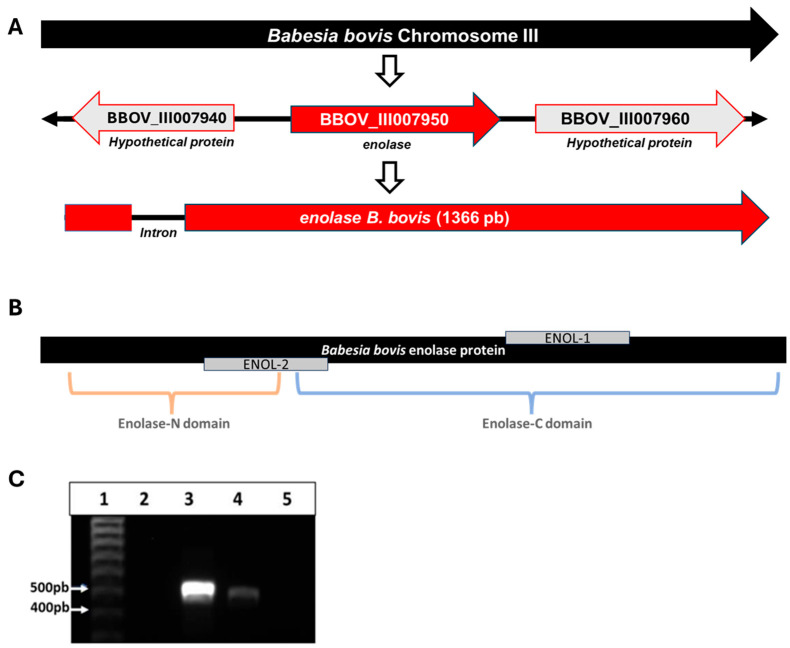
(**A**) Schematic representation of enolase gene location on *B. bovis* chromosome III. (**B**) Schematic representation of enolase protein with the two enolase domains identified by SMART and the location of peptides ENOL-1 and ENOL-2. (**C**) Amplicons obtained from the reverse transcription-PCR assay with specific primers for the enolase gene on 1% agarose gel stained with GelRed^TM^. (1) Molecular size marker, (2) negative reaction control, without DNA or cDNA, (3) *B. bovis* DNA, (4) cDNA from *B. bovis*, (5) RT negative control, RNA treated with DNase but without retro transcriptase.

**Figure 2 vaccines-13-00818-f002:**
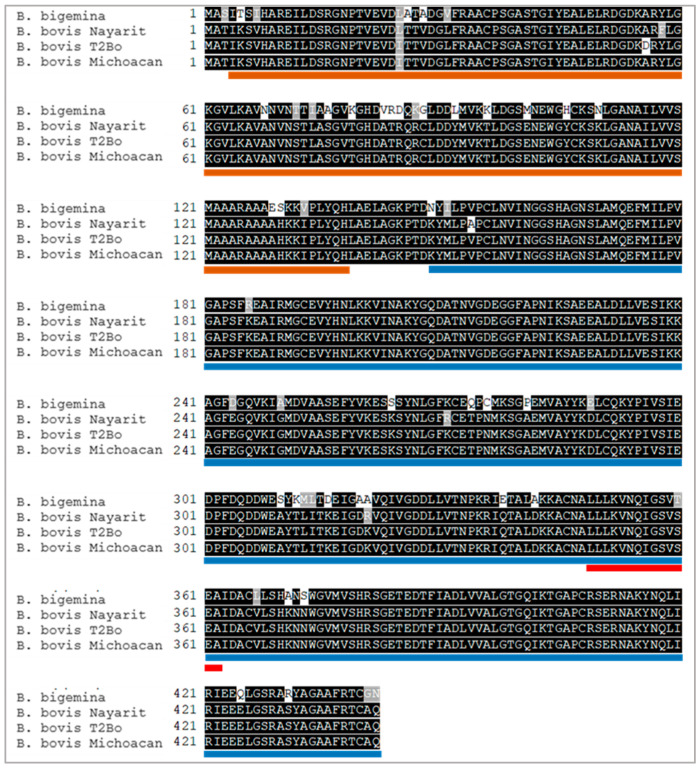
Multiple alignment of the *B. bovis* enolase predicted amino acid sequence. The sequences of Nayarit and Michoacan isolates were compared with the sequence obtained from the T2Bo sequence (AAXT00000000.2) and *B. bigemina* enolase (QEG79394.1). Orange and blue lines represent the Enolase-N and Enolase-C domains predicted by SMART. The red line represents the signature pattern of enolase identified by PROSITE.

**Figure 3 vaccines-13-00818-f003:**
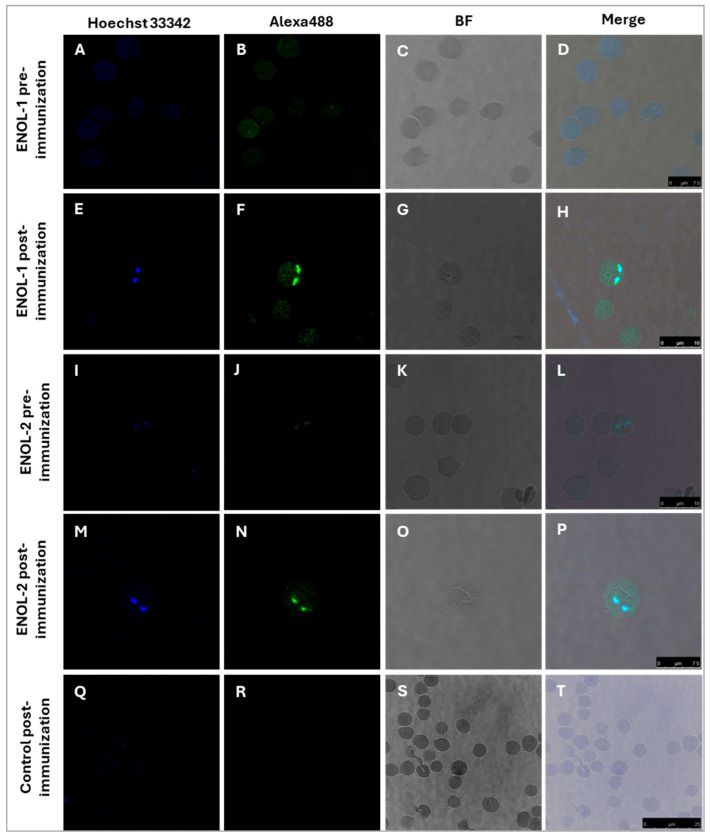
Enolase is expressed in *B. bovis* merozoites. IEP were incubated with sera from a rabbit immunized with peptide ENOL-1 (pre-immunization panels (**A**–**D**) and post-immunization panels (**E**–**H**)) and peptide ENOL-2 (pre-immunization panels (**I**–**L**) and post-immunization panels (**M**–**P**)). No signal was observed when merozoites were incubated with the post-immunization serum from a rabbit immunized with adjuvant (panels (**Q**–**T**)). Panels (**A**,**E**,**I**,**M**,**Q**) show the Hoechst 33342-stained nuclei. Panels (**B**,**F**,**J**,**N**,**R**) show the filter with Alexa Fluor 488. Panels (**C**,**G**,**K**,**O**,**S**), show the bright field images and panels (**D**,**H**,**L**,**P**,**T**), show the merged images. 100×. The black line indicates the micrometer ruler.

**Figure 4 vaccines-13-00818-f004:**
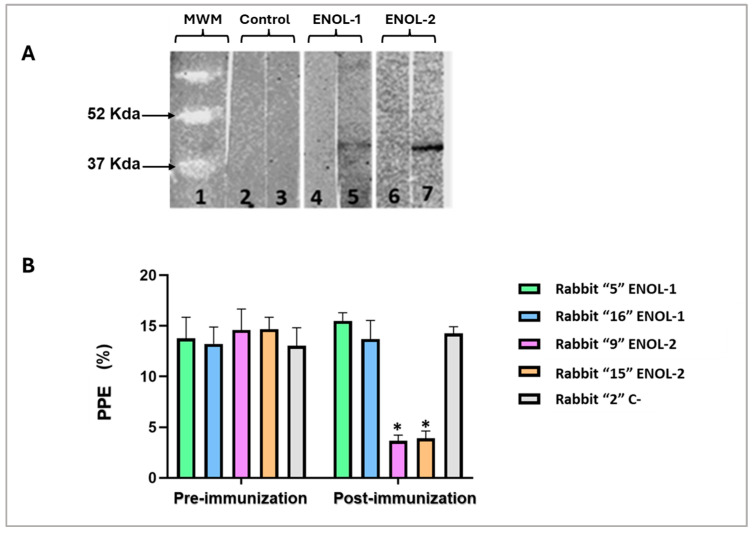
(**A**) The *B. bovis* enolase protein is recognized by anti-enolase antibodies by WB in *B. bovis*-infected red blood cells. Line 1: molecular weight marker; line 2: adjuvant control pre-immunization serum; line 3: adjuvant control post-immunization serum; line 4: ENOL-1 pre-immunization serum; line 5: ENOL-1 post-immunization serum; line 6: ENOL-2 pre-immunization serum; line 7: ENOL-2 post-immunization serum. (**B**) Neutralization Assay. The values on the Y–axis are shown as percentages of parasitized erythrocytes (PPE). On X–axis the results obtained from the evaluation of the inhibition generated by different antisera are shown. * = significant difference between the pre- and post-immunization serum samples (*p* < 0.05). Only sera from rabbits immunized with ENOL-2 showed significant statistical difference. The original Western blot figures can be found in Appendix A.

## Data Availability

The data presented in this study are available in this article and Appendix A.

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
