# Peer review of "Babesia bovis Enolase Is Expressed in Intracellular Merozoites and Contains B-Cell Epitopes That Induce Neutralizing Antibodies In Vitro"

_vaccines, 2025, doi:10.3390/vaccines13080818_

Round 1
Reviewer 1 Report
Comments and Suggestions for Authors
The authors have identified babesia Bovis protein Enolase that is expressed in intracellular merozoites, that is immunogenic in nature and the antibodies against which can neutralize the Babesia infection of erythrocytes. The reviewer has few queries
- Figure 3, the legend needs to be written on figure. On the left you can write pre and Post immunization sera as it is from ENOL-1 and 2. On the top write DAPI, anti-enolase instead of writing it all in text.
- ENOL-1 field shows two other cells that don’t display green fluorescence. What could be the reason?
- What about the expression of enolase during other stages of babesia?
- Is the antigen expressed on the erythrocyte surface? Can the authors perform a non-permeabilization IFA to establish if the enolase is expressed on the surface.
- The authors should also perform an IFA or western using sera from an infected animal to establish if the protein is exported out.
- The authors have checked whether the antibodies can neutralize Babesia and prevent its new infection? The authors should also check whether the antibodies can affect replication?
Author Response
Comment 1. Figure 3, the legend needs to be written on figure. On the left you can write pre and Post immunization sera as it is from ENOL-1 and 2. On the top write DAPI, anti-enolase instead of writing it all in text.
Response 1. Thank you for your suggestion. The modifications have been added to Figure 3. About the DAPI stain, that was our mistake, we did not use DAPI, we used Hoechst 33342 instead, and thanks to your observation that had been corrected in the figure and in the text.
Comment 2 . ENOL-1 field shows two other cells that don’t display green fluorescence. What could be the reason?
Response 2. In panel H, we observe three erythrocytes, just as we do in panel G. However, only one of these erythrocytes is infected with two B. bovis merozoites. We can confirm this because panel E shows only two small nuclei. The other two erythrocytes are not infected and therefore do not display green fluorescence.
Comment 3. What about the expression of enolase during other stages of babesia?
Response 3. Thank you for your question. The aim of this study was to characterize enolase expression specifically in the intraerythrocytic stages of Babesia bovis, which is why we chose the title of the manuscript. However, future research should focus on characterizing the expression and function of enolase in the parasite stages that infect the tick vector.
Comment 4. Is the antigen expressed on the erythrocyte surface? Can the authors perform a non-permeabilization IFA to establish if the enolase is expressed on the surface.
Response 4. We greatly appreciate the suggestion and agree that a non-permeabilization IFA assay would provide valuable information on the potential surface localization of enolase. However, the objective of this study was to demonstrate enolase expression in intraerythrocytic merozoites and to evaluate whether anti-enolase antibodies can neutralize Babesia bovis invasion of erythrocytes. This was aimed at assessing enolase as a potential vaccine candidate. Although surface localization studies were not part of the current work, they should definitely be performed in future studies to confirm the subcellular localization of enolase. We added a note in the Discussion section on the importance of establishing the subcellular and surface localization of enolase in future studies (lines 396-399)
Comment 5. The authors should also perform an IFA or western using sera from an infected animal to establish if the protein is exported out.
Response 5. Thank you for your suggestion. We agree that performing an assay to determine whether enolase is exported would provide valuable information. However, this was not the aim of the present study. One of the major challenges in evaluating exported proteins in Babesia bovis is that most of the infected erythrocytes do not circulate in the peripheral blood of infected cattle; instead, they are sequestered in the capillaries of major organs, which limits experimental approaches involving serum from infected animals. However, we added a note in the Discussion section on the importance of performing such assays in future studies (lines 396-399).
.
Comment 6. The authors have checked whether the antibodies can neutralize Babesia and prevent its new infection? The authors should also check whether the antibodies can affect replication?
Response 6. Thank you for your question and your suggestion. In the in vitro neutralization assay with anti-ENOL2 serum, we observed a 71.99% reduction in parasitemia, demonstrating that the antibodies effectively blocked merozoite entry into erythrocytes. Due to the parasite's replication cycle inside erythrocytes, which is about 3.5-5 cycles per day for B. bovis (https://doi.org/10.1016/s1286-4579(03)00041-8), and because this neutralization assay was evaluated for 72 h, we can speculate that this blockade of invasion prevents replication, as the multiplication cycle within the vertebrate host depends exclusively on the intraerythrocytic phase. However, whether anti-enolase antibodies block infection of the tick, therefore preventing new infection in the mammalian host, was not evaluated. This study did not include tick stages, as mentioned above. However, we added a note in the Discussion section on the importance of performing such assays in future studies (lines 413-415).

Reviewer 2 Report
Comments and Suggestions for Authors
Bovine babesiosis caused by the tick-borne apicomplexan parasite Babesia bovis is an economically important disease that threatens the cattle industry worldwide. The disease results in high mortality and morbidity and is characterized by anemia, fever, and sequestration of parasitized red blood cells in blood capillaries. Both strong innate and adaptive immune responses are an important component of protection against intraerythrocytic protozoan parasites. One approach to the vaccination of Bovine babesiosis, which is currently considered a priority, is the development of a recombinant vaccine that probably includes several protective antigens. This approach offers potential advantages over traditional live-attenuated vaccines, since recombinant vaccines are safe and do not cause disease.
In this study, the enolase gene of two B. bovis isolates was successfully identified and characterized through sequenced, cloning, and expression. In addition, two peptides containing predicted B-cell epitopes were determined in this study. Although both identified synthetic peptides were highly immunogenic when immunized in rabbits, only one of them likely contain a B-cell epitope that induce neutralizing antibodies and can be considered as a vaccine candidate. For this reason, the enolase is a candidate antigen of significant interest for vaccine development. However, it seems to me that one antigen will not be enough and other potential antigens that induce the development of a cellular immune response, inhibiting growth and invasion of the pathogen.
Although in my opinion the manuscript is interesting, the text should be significantly corrected for publication. I hope that further challenge-exposure with virulent B. bovis strain in immunized cattle will be carried out to confirm the authors' hypothesis.
The quality of the figure S-3 needs to be improved.
Author Response
Comment 1. Bovine babesiosis caused by the tick-borne apicomplexan parasite Babesia bovis is an economically important disease that threatens the cattle industry worldwide. The disease results in high mortality and morbidity and is characterized by anemia, fever, and sequestration of parasitized red blood cells in blood capillaries. Both strong innate and adaptive immune responses are an important component of protection against intraerythrocytic protozoan parasites. One approach to the vaccination of Bovine babesiosis, which is currently considered a priority, is the development of a recombinant vaccine that probably includes several protective antigens. This approach offers potential advantages over traditional live-attenuated vaccines, since recombinant vaccines are safe and do not cause disease.
In this study, the enolase gene of two B. bovis isolates was successfully identified and characterized through sequenced, cloning, and expression. In addition, two peptides containing predicted B-cell epitopes were determined in this study. Although both identified synthetic peptides were highly immunogenic when immunized in rabbits, only one of them likely contain a B-cell epitope that induce neutralizing antibodies and can be considered as a vaccine candidate. For this reason, the enolase is a candidate antigen of significant interest for vaccine development. However, it seems to me that one antigen will not be enough and other potential antigens that induce the development of a cellular immune response, inhibiting growth and invasion of the pathogen.
Although in my opinion the manuscript is interesting, the text should be significantly corrected for publication. I hope that further challenge-exposure with virulent B. bovis strain in immunized cattle will be carried out to confirm the authors' hypothesis.
Response 1. Thank you for your kind comments. Thanks to your comments and those of the other reviewers, the text has been significantly corrected, including the summary. In response to your comments, in lines 48-50 of the summary, we emphasize the need for further experimental studies to fully elucidate its functional role and validate its candidacy as a vaccine antigen.
Comment 2. The quality of the figure S-3 needs to be improved.
Response 2. The structure and quality of figure S-3 were modified according to your suggestion.

Reviewer 3 Report
Comments and Suggestions for Authors
- Introduction. The introduction is very well-founded. The authors provide convincing facts about the losses caused by this parasite. In Latin America, this disease is transmitted by the cattle fever ticks Rhipicephalus microplus and R. annulatus. Estimated annual economic losses caused by the cattle fever tick and the diseases they transmit, such as babesiosis and anaplasmosis, are between US$13.9 billion and US$18.7 billion. Developing effective and sustainable methods to control this disease should be an important goal for research in countries where bovine babesiosis is endemic. Since the late 20th century, efforts have been made to develop an effective vaccine against bovine babesiosis. This has led to the creation of vaccines using attenuated live parasites, which have proven very effective. However, their use comes with several drawbacks, including the risk of reversion to virulence, high production costs, and potential contamination with other pathogens, among others.
- Materials and Methods – The authors used 9 modern methods to achieve their goal. The methods are described in detail so that they can be reproduced in other laboratories.
- The results obtained are presented in 4 figures. Three additional figures are also presented, which provide additional information about the results obtained, namely:
- Figure S-1. A) Nucleotide multiple alignment of predicted B. bovis enolase, identified in Chromosome III with genomic sequence of T2Bo strain (AAXT01000001.1) with enolases from P. falciparum (XM_004221995.1) and B. bigemina (MK490919.1) obtained with Clustal W and edited with boxshade.
- Figure S-2. Antibody determination graphs against B. bovis enolase peptides.
- Figure S-3. Antibody titration graphs against B. bovis enolase peptides.
- The results obtained were analyzed very well by comparing them with the studies of other researchers. A total of 23 scientific publications were cited.
There is only one small grammatical error on line 85 where it is written: he enolase gene sequences of Babesia bigemina (NCBI accession code: MK490919.1) and…. I recommend that the authors correct it as follows: The enolase gene sequences of Babesia bigemina (NCBI accession code: MK490919.1) and………
Author Response
Comment 1.
- Introduction. The introduction is very well-founded. The authors provide convincing facts about the losses caused by this parasite. In Latin America, this disease is transmitted by the cattle fever ticks Rhipicephalus microplus and R. annulatus. Estimated annual economic losses caused by the cattle fever tick and the diseases they transmit, such as babesiosis and anaplasmosis, are between US$13.9 billion and US$18.7 billion. Developing effective and sustainable methods to control this disease should be an important goal for research in countries where bovine babesiosis is endemic. Since the late 20th century, efforts have been made to develop an effective vaccine against bovine babesiosis. This has led to the creation of vaccines using attenuated live parasites, which have proven very effective. However, their use comes with several drawbacks, including the risk of reversion to virulence, high production costs, and potential contamination with other pathogens, among others.
- Materials and Methods – The authors used 9 modern methods to achieve their goal. The methods are described in detail so that they can be reproduced in other laboratories.
- The results obtained are presented in 4 figures. Three additional figures are also presented, which provide additional information about the results obtained, namely:
- Figure S-1. A) Nucleotide multiple alignment of predicted B. bovis enolase, identified in Chromosome III with genomic sequence of T2Bo strain (AAXT01000001.1) with enolases from P. falciparum (XM_004221995.1) and B. bigemina (MK490919.1) obtained with Clustal W and edited with boxshade.
- Figure S-2. Antibody determination graphs against B. bovis enolase peptides.
- Figure S-3. Antibody titration graphs against B. bovis enolase peptides.
- The results obtained were analyzed very well by comparing them with the studies of other researchers. A total of 23 scientific publications were cited.
There is only one small grammatical error on line 85 where it is written: he enolase gene sequences of Babesia bigemina (NCBI accession code: MK490919.1) and…. I recommend that the authors correct it as follows: The enolase gene sequences of Babesia bigemina (NCBI accession code: MK490919.1) and………
Response 1. Thank you for your comments. The correction suggested in line 85, now line 92, is already made.

Reviewer 4 Report
Comments and Suggestions for Authors
The concept and purpose of the paper are clearly defined by the authors. The title could be edited with deletion of the word "predicted" as neutralizing antibodies were detected in rabbits. To be more precise adding the phrase "induce neutralizing antibodies in vitro" is a suggestion. There could be more discussion about next steps for evaluating the ENOL-2 as a vaccine candidate targeting ruminants.
The conclusions are not overstated, and the paper requires light editing.
Line 54: Developing effective...; Line 80: erythrocytes; Line 85: Using the...;
Line 116: establish; Line 242: into; Line 246: insert missing information; Figure 1: chromosome is misspelled; Figure 3 could be improved with a column on the left with labels for each row (Pre-ENOL-1, Post ENOL-1; Pre-ENOL-2; Post-ENOL-2; Control. Figure 4 would improve with labels on the gel image at the top (Marker; Control; ENOL-1; ENOL-2)
Author Response
Comment 1. The concept and purpose of the paper are clearly defined by the authors. The title could be edited with deletion of the word "predicted" as neutralizing antibodies were detected in rabbits. To be more precise adding the phrase "induce neutralizing antibodies in vitro" is a suggestion.
Response 1. Thank you for your suggestion. We agree with your comment that the word “predicted” is not necessary in the title, and we have added the term “in vitro” for precision. The revised title is now: Babesia bovis enolase is expressed in intracellular merozoites and contains B-cell epitopes that induce neutralizing antibodies in vitro.
Comment 2. There could be more discussion about next steps for evaluating the ENOL-2 as a vaccine candidate targeting ruminants.
Response 2. Thank you for your observation, we have added more discussion about next steps evaluating ENOL-2 in lines 409-426.
Comment 3. The conclusions are not overstated, and the paper requires light editing.
Response 3. Thanks to your comments and those of the other reviewers, the text has been significantly corrected, for this comment, the edited conclusions section now runs from lines 418 to 428.
Comment 4. Line 54: Developing effective...;
Response 4. Now line 61 corrected as mentioned.
Comment 5. Line 80: erythrocytes;
Response 5. Now line 86 corrected as mentioned
Comment 6. Line 85: Using the...;
Response 6. The sentence has been modified according to yours and another reviewer’s suggestion. Now in line 92.
Comment 7. Line 116: establish;
Response 7. Now in line 124 the correction has been made.
Comment 8 Line 242: into;
Response 8. Now line 250, it was corrected as mentioned.
Comment 9 Line 246: insert missing information;
Response 9. Now line 252 corrected and added to bibliography.
Comment 10 Figure 1: chromosome is misspelled;
Response 10. Figure 1was edited and chromosome correctly spelled. Thank you.
Comment 11. Figure 3 could be improved with a column on the left with labels for each row (Pre-ENOL-1, Post ENOL-1; Pre-ENOL-2; Post-ENOL-2; Control.
Response 11. Figure 3 was modified thank you to your suggestion and that of another Reviewer.
Comment 12. Figure 4 would improve with labels on the gel image at the top (Marker; Control; ENOL-1; ENOL-2).
Response 12. Figure 4 is now edited. We appreciate this suggestion.
Thank you for all your corrections. We have carefully addressed each of them and now the text has improved considerably.

Round 2
Reviewer 1 Report
Comments and Suggestions for Authors
The manuscript has improved considerably after revisions.
Author Response
We thank the reviewer.